# Third-Order Effective Properties for Random-Packing Systems Using Statistical Micromechanics Based on a GPU Parallel Algorithm in Fast Computing *n*-Point Correlation Functions

**DOI:** 10.3390/ma15165799

**Published:** 2022-08-22

**Authors:** Shaobo Sun, Huisu Chen, Jianjun Lin

**Affiliations:** 1Jiangsu Key Laboratory of Construction Materials, School of Materials Science and Engineering, Southeast University, Nanjing 211189, China; 2Key Laboratory of Green Construction and Intelligent Maintenance for Civil Engineering of Hebei Province, School of Civil Engineering & Mechanics, Yanshan University, Qinhuangdao 066004, China

**Keywords:** *n*-point correlation function, third-order approximation, effective thermal conductivity, effective diffusion coefficient, shape effect

## Abstract

Estimating the effective properties of a particulate system is the most direct way to understand its macroscopic performance. In this work, we accurately evaluate the third-order approximations involving the three-point microstructural parameter ζ, which can be calculated from a triple integral involving 1-, 2-, and 3-point correlation functions. A GPU-based parallel algorithm was developed for quickly computing the *n*-point correlation functions, and the results agree well with analytical solutions. The effective thermal conductivity and diffusion coefficient are calculated by the third-order approximates for the random-packing systems of a super-ellipsoid. By changing the parameters of the super-ellipsoid, the particle-shape effect can be predicted for both the thermal conductivity and diffusion coefficient.

## 1. Introduction

The quantitative characterization of heterogeneous materials from their microstructure is a long-studied issue with applications in many scientific fields [1,2,3,4,5,6]. For a particulate system, the most direct way to understand its macroscopic performance is by estimating its effective properties [7]. According to the composite material theory, however, an exact analytical prediction model never exists even for the simplest class of transport processes, such as thermal conduction and diffusion, because an infinite amount of microstructural information is required in the form of *n*-point correlation functions. Bounds or prediction models involving sole 1-point correlation function, also known as the packing fraction or volume fraction, can only provide a relatively coarse estimation, i.e., the well-known Hashin–Shtrikman (H–S) bounds [8,9]. To improve these bounds, additional information on the statistics of the spatial distribution of the phases is needed.

Brown [10] first brought higher-order parameters to effective properties. Based on H–S bounds, Beran [11] derived third-order bounds for effective thermal conductivity using variational principles in a homogeneous and isotropic system, but the results are complicated due to their involvement of sixfold integrals. Torquato [12] and Milton [13] independently simplified these complex bounds and their formulas are expressed by a volume fraction, ϕ, and a third-order microstructural parameter, ζ, which is a triple integral involving 1-, 2-, and 3-point correlation functions. In particular, for an isotropic two-phase system, Torquato [14] derived a three-point approximation formula that has shown a good agreement with simulation data [15]. Therefore, with this parameter, ζ, higher-order bounds or approximations can be obtained directly.

Despite decades of progress in computer technology, a direct computation of the microstructural parameter ζ remains a challenge. Most works involving ζ have somehow bypassed the direct computation of this higher-order parameter [16,17,18,19], until Gillman and Matouš [20] proposed adaptive interpolation methods to reduce the complexity of the three-point correlation function, and the results were verified by a penetrable sphere model [21]; later, Gillman et al. [22] extended their work and another three-point parameter η was computed with high accuracy. In their work, 500–7200 CPU computing cores are employed. Hlushkou et al. [23] employed a non-uniform spatial grid for the construction of a sampling template to reduce computation cost because the region of integration near zero makes a substantial contribution to the integral. The results were verified with face-centered cubic (FCC) and body-centered cubic (BCC) sphere arrays by analytical results [24]. Li [25] computed the third-point parameter with different volume fractions of glass spheres, but this yielded the results of simple 2D slices. Therefore, the computation cost of the parameter remains enormous; an ordinary algorithm based on a CPU is not enough to study the effective properties of more complex particulate systems.

In this paper, by employing the non-uniform spatial grid-sampling method [23], a GPU-based parallel algorithm is developed to compute the microstructural parameter, ζ, and an effective thermal conductivity and diffusion coefficient for packing systems of super-ellipsoids is are calculated according to Torquato’s third-order approximation [14]. The rest of this paper is organized as follows: Section 2 describes the fundamental theories; Section 2.1 describes the *n*-point correlation functions, including the standard form of 1-, 2-, and 3-point correlation functions; Section 2.2 details the effective material behavior, including the thermal conductivity and diffusion coefficient; in Section 3, we provide the details of a GPU-based parallel algorithm in computing the microstructural parameter ζ and its verifications; and the results and conclusions are presented in Section 4 and Section 5.

## 2. Fundamental Theory

This section provides a basic description of *n*-point correlation functions and an overview of the higher-order statistical micromechanics for calculating the effective thermal conductivity and diffusion coefficient of statistically isotropic microstructures.

### 2.1. n-Point Correlation Functions

The *n*-point correlation function is a generalized definition of correlation functions. In this work, only standard one-, two-, and three-point correlation functions, which are included in the higher-order parameter ζ, are considered. The *n*-point correlation function is defined as the probability that *n* points with locations $x_{1},x_{2},\ldots ,x_{n}$ will be in the same phase i:(1)$$S_{n}^{\left(i\right)}\left(x_{1},x_{2},\ldots ,x_{n}\right)=P\left\{I^{\left(i\right)}\left(x_{1}\right)=1,I^{\left(i\right)}\left(x_{2}\right)=1,\ldots ,I^{\left(i\right)}\left(x_{n}\right)=1\;\right\},$$
it can be expressed as expectation (or average) of the multiplication of indicator functions at the *n* locations
(2)$$S_{n}^{\left(i\right)}\left(x_{1},x_{2},\ldots ,x_{n}\right)=\langle I^{\left(i\right)}\left(x_{1}\right)I^{\left(i\right)}\left(x_{2}\right)\cdots I^{\left(i\right)}\left(x_{n}\right)\rangle$$
where the angular bracket 〈⋯〉 denotes the expectation or ensemble average, and $I^{\left(i\right)}$ represent the indicator function for phase i,
(3)$$I^{\left(i\right)}\left(x\right)=\left\{\begin{array}{c} 1,\;x\in V_{i},\\ 0,\;x\in \bar{V_{i}}, \end{array}\right.$$
where $V_{i}$ is the region occupied by phase i with packing fraction $\phi_{i}$. Therefore, one can define one-, two-, and three-point correlation functions when *n* = 1, 2, and 3,
(4)$$S_{1}^{\left(i\right)}\left(x\right)=P\left\{I^{\left(i\right)}\left(x\right)=1\;\right\}=\langle I^{\left(i\right)}\left(x\right)\rangle$$
(5)$$S_{2}^{\left(i\right)}\left(x_{1},x_{2}\right)=P\left\{I^{\left(i\right)}\left(x_{1}\right)=1,I^{\left(i\right)}\left(x_{2}\right)=1\;\right\}=\langle I^{\left(i\right)}\left(x_{1},x_{2}\right)\rangle$$
(6)$$S_{3}^{\left(i\right)}\left(x_{1},x_{2},x_{3}\right)=P\left\{I^{\left(i\right)}\left(x_{1}\right)=1,I^{\left(i\right)}\left(x_{2}\right)=1,I^{\left(i\right)}\left(x_{3}\right)=1\;\right\}=\langle I^{\left(i\right)}\left(x_{1},x_{2},x_{3}\right)\rangle$$

Considering a two-phase composite material (i.e., concrete), the gray phase represents “aggregate” in this work, and the white phase is a matrix. A schematic representation of one-, two-, and three-point correlation functions are shown in Figure 1, where $S_{aa}$ is another form of $S_{2}^{\left(i\right)}\left(r\right)$ when i represents an aggregate phase; in addition, $S_{mm}$ means the phase i of interest is a matrix. There is another function, $S_{am}$, which means one point lands on the aggregate, and the other point lands in the matrix, it is obvious that:
(7)$$S_{aa}+S_{mm}+S_{am}=1.$$

Similarly, $S_{aaa}$ means $S_{3}^{\left(i\right)}\left(r_{1},r_{2},\theta \right)$ when phase i of interest is aggregate:(8)$$S_{aaa}+S_{aam}+S_{amm}+S_{mmm}=1$$

In addition, limiting values of $S_{2}^{\left(i\right)}\left(r\right)$ can be obtained when r=0 and r=∞:(9)$$S_{2}^{\left(i\right)}\left(0\right)=\phi_{i},$$
(10)$$\underset{r\to \infty }{\lim}S_{2}^{\left(i\right)}\left(r\right)=\phi_{i}^{2}.$$

Similarly, limiting values of $S_{3}^{\left(i\right)}\left(r_{1},r_{2},\theta \right)$ can be obtained when $r_{1},r_{2}\to 0$ and $r_{1},r_{2}\to \infty ,$ θ≠0:(11)$$\underset{r_{1},r_{2}\to 0}{\lim}S_{3}^{\left(i\right)}\left(r_{1},r_{2},\theta \right)=\phi_{i}$$
(12)$$\underset{r_{1},r_{2}\to \infty ,\theta \neq 0}{\lim}S_{3}^{\left(i\right)}\left(r_{1},r_{2},\theta \right)=\phi_{i}^{3}$$

### 2.2. Third-Order Models of Effective Material Behavior

For isotropic two-phase materials, the well-known H–S bounds are generally regarded as the best second order bounds on effective properties such as magnetic permeability and electrical or heat conductivity and diffusivity, because of their mathematical equivalents. The H–S lower bound for effective thermal conductivity, $\kappa_{e},$ is expressed as follows:(13)$$\;\kappa_{e}=\kappa_{m}+\frac{1-\phi_{a}}{\frac{1}{\kappa_{a}-\kappa_{m}}+\frac{1-\phi_{a}}{3\kappa_{m}}}$$

The effective property $\kappa_{e}$ is dependent on the conductivity of matrix $\kappa_{m}$, the conductivity of the aggregate phase $\kappa_{a},$ and the volume fraction of aggregate $\phi_{a}$. Beran [11] improved the H–S bounds and derived the third-order bounds of $\kappa_{e}$, which was then simplified by Torquato [12] and Milton [13] independently, the lower bound can be expressed as:(14)$$\kappa_{e}=\phi_{a}\kappa_{a}+\phi_{m}\kappa_{m}-\frac{\phi_{a}\phi_{m}{\left(\kappa_{a}-\kappa_{m}\right)}^{2}}{\kappa_{v}\kappa_{a}+\phi_{a}\kappa_{m}+2{\left(\zeta_{a}/\kappa_{a}+\zeta_{m}/\kappa_{m}\right)}^{-1}}$$

For isotropic two-phase systems, Torquato [14] derived a three-point approximation that has shown good agreement with simulations for packing systems without large clustering. This third-order approximation is given as:(15)$$\frac{\kappa_{e}}{\kappa_{m}}=\frac{1+2\phi_{a}\beta_{am}-2\phi_{m}\zeta_{a}\beta_{am}^{2}}{1-\phi_{a}\beta_{am}-2\phi_{m}\zeta_{a}\beta_{am}^{2}}$$
where
(16)$$\beta_{am}=\frac{\kappa_{a}-\kappa_{m}}{\kappa_{a}+2\kappa_{m}}$$

In this work, a strong contrast ratio between particle and matrix phases is assumed for effective thermal conductivity, i.e.,
(17)$$\frac{\kappa_{a}}{\kappa_{m}}\approx \infty$$

Then, the equation (15) yields:(18)$$\frac{\kappa_{e}}{\kappa_{m}}=\frac{1+2\phi_{a}-2\phi_{m}\zeta_{a}}{1-\phi_{a}-2\phi_{m}\zeta_{a}}$$

In these formulas, the third-order microstructural parameter $\zeta_{i}\;\left(i=a,m\right)$ is introduced, involving 1-, 2-, and 3-point correlation functions. For 2D media:(19)$$\;\zeta_{i}=\frac{4}{\pi \phi_{a}\phi_{m}}\int_{0}^{\infty }\int_{0}^{\infty }\int_{0}^{\pi }\frac{1}{r_{1}r_{2}}\theta \cdot \cos\left(2\theta \right)\cdot {\tilde{S}}_{aaa}\left(r_{1},r_{2},\theta \right)d\left(\theta \right)dr_{1}dr_{2}$$

For 3D media:(20)$$\;\zeta_{i}=\frac{9}{2\phi_{a}\phi_{m}}\int_{0}^{\infty }\int_{0}^{\infty }\int_{-1}^{1}\frac{1}{r_{1}r_{2}}P_{2}\left(cos\theta \right)\cdot {\tilde{S}}_{aaa}\left(r_{1},r_{2},\theta \right)d\left(cos\theta \right)dr_{1}dr_{2}\;$$

Based on the third-order approximate formula, the effective diffusion coefficient $D_{e}$ in the packing system can be evaluated also assuming a strong contrast ratio [19,26], i.e.,
(21)$$\frac{D_{m}}{D_{a}}=\infty$$

In addition, for a two-phase material such as concrete, the aggregate phase contributes nothing to diffusion, and the matrix phase of the material (i.e., mortar) must be attributed to the whole effective property; therefore, the third-order approximation in terms of diffusion coefficient $D_{e}$ can be expressed as:(22)$$\frac{D_{e}}{D_{m}}\approx \frac{1}{1-\phi_{a}}\frac{1-\phi_{a}-0.5\left(1-\phi_{a}\right)\zeta_{a}}{1+0.5\phi_{a}-0.5\left(1-\phi_{a}\right)\zeta_{a}}.\;$$

## 3. GPU-Based Parallel Algorithm

For particulate packing systems, a GPU-based parallel packing algorithm is developed and described in this section. The periodic samplings method is proposed to match the periodic packing system. The efficiency of this algorithm is compared with the results of the CPU. In addition, the results are compared with analytical solutions of a penetrable sphere packing system and the existing results of an impenetrable sphere packing system, which verified the algorithm.

### 3.1. Sampling of n-Point (n = 1, 2, and 3) Correlation Functions

The three-point microstructural parameter ζ, for both 2D and 3D systems, can be seen as a triple integral involving 1-, 2-, and 3-point correlation functions. The 1-point correlation function is the volume fraction $\phi_{i}$ of the phase i. The 2-point correlation function is computed for each discrete value of r, according to the range of integration; theoretically, it has to be calculated from zero to infinity. The major challenge is the computation of 3-point correlation functions, because not only $r_{1}$ and $r_{2}$ have the integration range of zero to infinity, but also the angle θ between $r_{1}$ and $r_{2}$ must be considered $\theta =\left[0,\pi \right]$. It can be observed from the formula that the region of integration near $r_{1}=0$ and $r_{2}=0$ makes an enormous contribution to the integral due to the existence of $1/r_{1}$ and $1/r_{2}$—this will require a fine spatial grid in order to obtain an accurate result. However, a large number of discrete values of 2- and 3-point correlations will result in a huge computation cost.

Following the sampling method by Hlushkou et al. [23], for r, $r_{1}$, and $r_{2}$ the non-uniform sampling method is employed to reduce the discrete sampling number:(23)$$r_{n}=a_{0}D_{eq}A^{n}$$
where $D_{eq}$ is the equivalent diameter of a particle; $a_{0}$ is a constant near zero, determining the smallest r, $r_{1}$, and $r_{2}$; and A is a constant slightly larger than one, i.e., in this work, we take A=1.11.

All particle-packing systems considered in this study are employed in periodic boundary conditions. Thus, the periodic sampling method is proposed in this work to match the periodic-packing system. For $S_{2}^{\left(i\right)}\left(r\right)$ and $S_{3}^{\left(i\right)}\left(r_{1},r_{2},\theta \right)$, the start points $P_{1}$ of a sampling line or sampling triangle are random or uniformly distributed in the system. Therefore, there are chances that the end points of line segment $P_{2}$ or other vertices of a triangle $P_{2}$ and $P_{3}$ will fall outside the box. If the sampling lines or triangles are not allowed to fall outside box, the “boundary effect” will emerge, as shown in Figure 2a. On the contrary, when the sampling line or triangle is much larger than the packing system, they will not characterize the packing system correctly, as shown in Figure 2b. In order to solve these conflicting problems, the end points must be placed in the periodic position, as can be seen in the 2D examples from Figure 3a,b. The combination of the periodic-particle-packing system and the periodic sampling method allows us to characterize approximately infinite systems.

Although the non-uniform sampling methods have been employed, the cost of computation remains enormous, especially for the 3-point correlation function. For example, after a convergence study in this work, the range of discrete values for $r_{1}$ or $r_{2}$ is chosen from $10^{-15}D_{eq}$ to $15D_{eq}$, the constant A=1.11, θ is divided evenly into 100 parts, resulting in patterns of the sampling triangle $N_{pattern}=N_{r_{1}}\times N_{r_{2}}\times N_{\theta }=357\times 357\times 100=12,744,900$, and for each pattern, $N_{sample}=2\times 10^{6}$ are needed. It can be observed that the computation progress for each sampling triangle is highly parallelizable; Gillman et al. [22] used 500–7200 CPU cores to perform such challenging work. However, each computer has a limited number of CPU cores, i.e., intel Core i9-12900KS has only 16 CPU Cores; in contrast, a GPU is composed of thousands or millions of cores that can manage multiple threads simultaneously. In this study, a GPU-based parallel algorithm is developed to accelerate the computation.

The flowchart of the GPU parallel algorithm for computing the *n*-point correlation function is shown in Figure 4. Firstly, the particle information in the system is loaded, including the boundary condition (i.e., periodic in this work), the particle coordinate locations, the particle size, and other parameters (i.e., Euler angles). Then, sampling points/lines/triangles are generated, and the points are randomly or uniformly generated in the system; consequently, the end points of the lines and triangles are generated using the Monte Carlo method following the periodic conditions mentioned above. Then, we execute kernels in parallel with the GPU; when a kernel is launched, a grid of threads that are organized in a 3D hierarchy are generated, each gird is organized into an array of thread blocks, and each block can contain up to 1024 threads. Then, we allocate threads in advance according to the complexity of the computation; for example, for 1- or 2-point correlation functions, a 3D block and 3D grid are more suitable because the computing cores (threads) number in the millions, but for the 3-point correlation function, a 2D block and 2D grid are appropriate. However, in this work we use the 3D block and 3D grid to allocate the threads, because they can be degraded to the 2D block and 2D grid. This information is then input into the GPU kernels and the sampling points/lines/triangles are allocated to threads cores to execute judgement of the location and classification, the 1-point correlation function yields the result directly, and $S_{2}^{\left(i\right)}\left(r\right)$ and $S_{3}^{\left(i\right)}\left(r_{1},r_{2},\theta \right)$ will proceed to the next sample line/triangle, until all the patterns of the sample are calculated. Finally, all the discrete data are stored. To show the speedup ratio of the GPU to the CPU, an ordinary computer is used, i.e., the CPU is an AMD RYZEN 7 3800× and the “GPU” is NVIDIA GeForce RTX 2070. The results are listed in Table 1.

### 3.2. Verification

Berryman [27] derived an analytical expression for the *n*-point correlation functions of phase i using a fully penetrable sphere-packing model, which is one of the few packing systems for which the *n*-point correlation functions are defined analytically.
(24)$$S_{n}^{\left(i\right)}\left(x_{1},x_{2},\ldots ,x_{n}\right)=\exp\left\{-\rho V_{n}\right\}.$$

For 2-point correlation function of phase i,
(25)$$S_{2}^{\left(i\right)}\left(r\right)=\exp\left\{-\rho V_{2}\right\},\;$$
where ρ is the number density of the spheres, and $V_{n}$ represents the union volume of n spheres, when n=2,
(26)$$\frac{V_{2}\left(rR\right)}{R^{3}}=\left\{\begin{array}{c} \frac{4\pi }{3}\left(1+\frac{3}{4}r-\frac{r^{3}}{16}\right),\;r\leq 2\\ \frac{8\pi }{3},\;\;\;\;\;\;\;\;\;\;\;\;\;\;\;\;\;\;\;\;\;\;\;\;\;r\geq 2 \end{array}\right.\;$$
where R is the sphere radius.

Figure 5a shows the comparison of the analytical and estimated 2-point correlation function of the matrix phase for the penetrable sphere packing. It is evident that the algorithm is able to calculate the 2-point correlation function with a high accuracy. In addition, the 2-point correlation function holds the relationships $S_{mm}\left(0\right)=0.5=\phi_{m}$ and $\underset{r\to \infty }{\lim}S_{mm}\left(r\right)=0.25=\phi_{m}^{2}$, thereby satisfying the limiting value of $S_{2}^{\left(i\right)}\left(r\right)$. Figure 5b shows the results of the $\zeta_{m}$ of the matrix phase for penetrable sphere packing and said results compared with other results in the literature [20,21]. This verified our algorithm in penetrable packing systems. As the packing of an impenetrable sphere model has been studied extensively [16,17,18,19,20], to better verify our algorithm, an impenetrable packing system of a monodisperse sphere was generated using a DEM-based method [28] to ensure the statistical isotropy; after a convergence study, the container side was set to be 20 times the length of the particle diameter (i.e., L/D=20), and the results of the impenetrable sphere packing system are shown in Figure 6a. Figure 6b is also shown to ensure that there is not any overlap between the particles. Impenetrable packing spheres with a packing fraction of $\phi_{a}=0.1-0.6$ were calculated using our approach, and the results were compared with existing ones, as shown in Figure 6c. An excellent agreement is shown between our results and the latest results by Gillman [20] and the simulation results by Chan and Torquato [15]. Therefore, the results of both penetrable and impenetrable packing systems verified our approach.

## 4. Results and Discussion

With the 3-point microstructural parameter ζ accurately estimated, the effective properties of thermal conductivity, $\kappa_{e}$, and the diffusion coefficient $D_{e}$ can be calculated for almost any packing system. In this work, super-ellipsoid was employed due to its morphological diversity, it is a joint name used to define a variety of 3D regular-shaped particles including ellipsoids, cuboids, etc. In mathematics, it can be expressed by adding a deformation coefficient m to the standard formula of an ellipsoid,
(27)$$\frac{{\left|x\right|}^{2m}}{a^{2m}}+\frac{{\left|y\right|}^{2m}}{b^{2m}}+\frac{{\left|z\right|}^{2m}}{c^{2m}}=1.$$
where a, b, and c denote the three semi-axes in coordinates.

Table 2 displays a fraction of the super-ellipsoid family with different values of the aspect ratio a/b and deformation coefficient m, in which b=c and m≥1. By changing the aspect ratio and deformation coefficient, we can study the effect of particle shape in two directions.

Figure 7a presents the third-order approximations of thermal conductivity $\kappa_{e}$ using Equation (18) for different packing fractions. First, the effect of the aspect ratio is explored. It can be observed that as the packing fracture $\phi_{a}$ increases, the effect of the particle shape becomes more pronounced; for example, the normalized effective thermal conductivity ($\kappa_{e}/\kappa_{m}$) for spheres (a=b=c) and ellipsoids (a/b=2.5,m=1, b=c) at packing fraction $\phi_{a}=0.1$ are basically the same, while at the packing fraction $\phi_{a}=0.5$, the effective thermal conductivity for an ellipsoid (a/b=2.5,m=1, b=c) is almost two times as high as the value for spheres. In general, “long” particles (i.e., a high aspect ratio ellipsoid) lead to a greater thermo conductivity. These results are also compared to the H–S lower bound, which is unable to characterize any shape effect. Then, we studied the effect of the deformation coefficient on the thermo conductivity; when m is changes from one to infinity, the super-ellipsoid (i.e., a=b=c) transforms from a sphere into a regular hexahedron, as shown in Figure 7b. With the increase of the packing fraction, the shape effect becomes more and more obvious, but the overall magnitude is less pronounced.

The third-order approximations of the diffusion coefficient $D_{e}$ were calculated for different packing systems using equation (22). Figure 7c shows the normalized $D_{e}$ as a function of the packing fraction. Unlike the effective thermal conductivity, the effective diffusion coefficient in all the packing systems decreases with the increase of the packing fraction. This is because the increasing particle phase will obstruct the diffusion. For the aspect ratio’s effect, a clear shape effect was also discovered, which becomes most obvious when the packing fraction $\phi_{a}$ is around 0.3–0.4. Overall, the packing of the particles with a high aspect ratio will more rapidly decrease the $D_{e}$. These results are also compared with the H–S lower bounds; it should be noted that some packing systems (i.e., a/b=2.5,a/b=2.0, b=c) in a range of packing fractions (i.e., $\phi_{a}=0\sim 0.3$) will exceed the H–S lower bounds. Figure 7d represents the effect of the deformation coefficient; it shows a similar tendency, but with a smaller magnitude.

## 5. Conclusions

In this work, third-order effective properties including thermal conductivity and the diffusion coefficient were estimated with a high accuracy and efficiency. The GPU-based parallel algorithm was developed to calculate the 3-point parameter. Verifications were conducted for both the penetrable and impenetrable sphere packing systems. The results of the 2-point correlation function and the 3-point parameter were verified by analytical expressions and recent simulation results. The effective thermo conductivity of the impenetrable sphere packing system was compared to the existing results and showed a good agreement.

With our GPU-based parallel algorithm, we calculated third-order approximations of thermal conductivity and a diffusion coefficient for a differently shaped super-ellipsoids packing system with a different packing fraction. A significant particle shape effect was predicted for both the thermal conductivity and diffusion coefficient, especially when increasing the aspect ratio of the particle. In addition, a clear shape effect could be observed when the particle transformed from a sphere to a regular hexahedron, but the magnitude was less pronounced. With these techniques, effective properties for a wide range of particle-packing systems can be predicted with high accuracy and speed.

## Figures and Tables

**Figure 1 materials-15-05799-f001:**
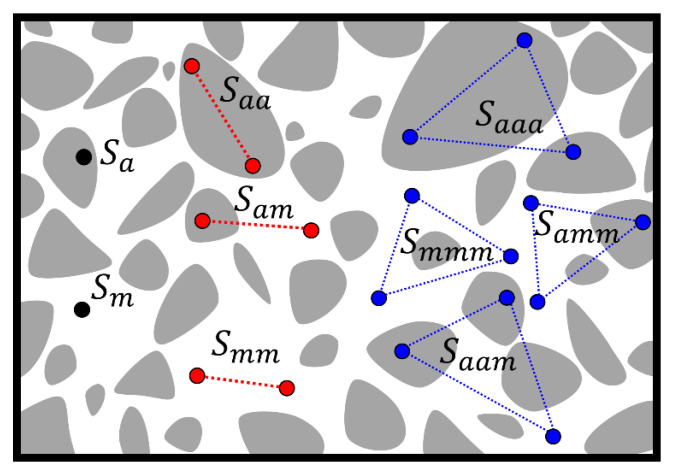
Illustration of one-, two-, and three-point correlation functions.

**Figure 2 materials-15-05799-f002:**
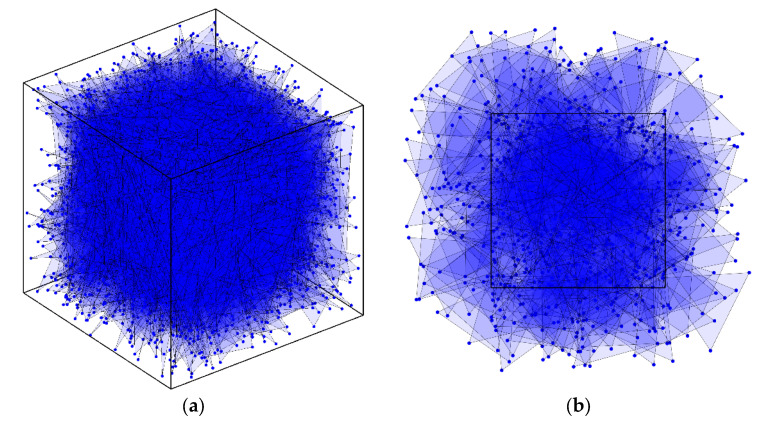
Examples of inaccurate sampling: (**a**) Sampling triangles are selected when all end points end in the container (boundary effect); (**b**) End points are randomly generated in the space, exceeding the system (2D illustration for a clear view).

**Figure 3 materials-15-05799-f003:**
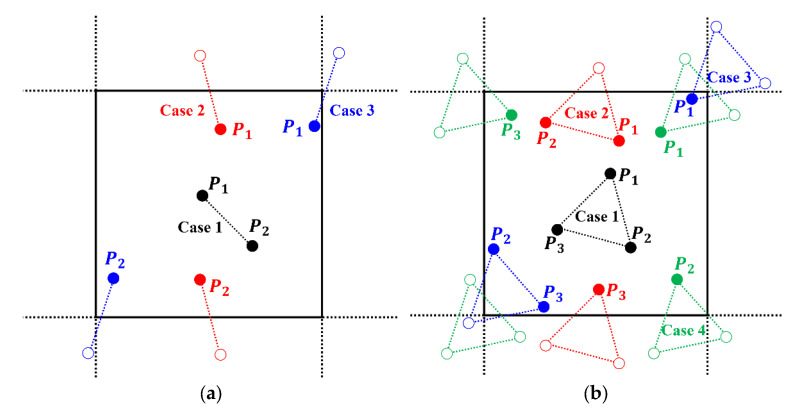
Illustration of periodic sampling method for 2D system: (**a**) Three cases of 2-point sampling; (**b**) Four cases of 3-point sampling.

**Figure 4 materials-15-05799-f004:**
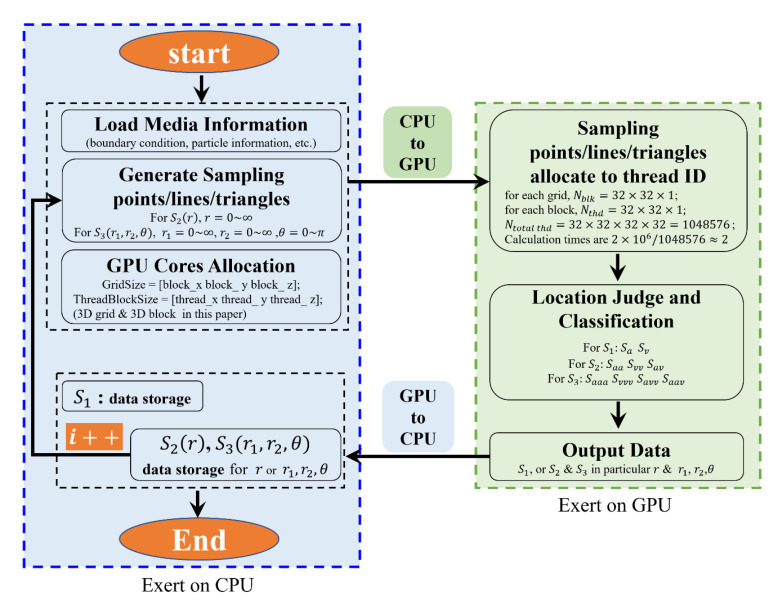
Flowchart of GPU-based parallel computing algorithm.

**Figure 5 materials-15-05799-f005:**
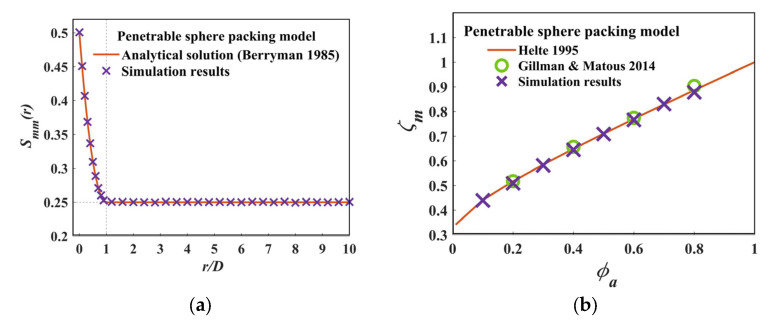
(**a**) Comparison of $S_{mm}$ in this work and analytical solutions [27]; (**b**) Comparison of $\zeta_{m}$ in this work with existing results [20,21].

**Figure 6 materials-15-05799-f006:**
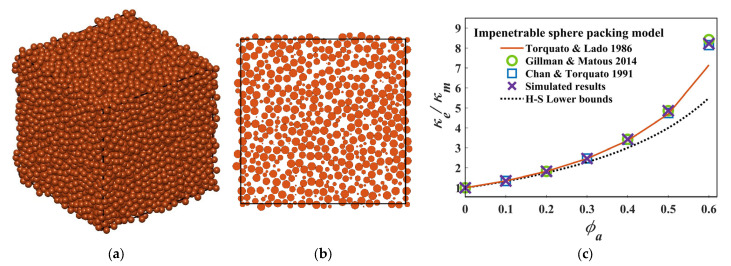
(**a**) An impenetrable sphere packing system, with a packing fraction $\phi_{a}=0.6$; (**b**) a section of the packing system; (**c**) comparison of the results of normalized thermal conductivity of third-order approximation for impenetrable sphere packing systems $\phi_{a}=0.1-0.6$ [8,15,17,20].

**Figure 7 materials-15-05799-f007:**
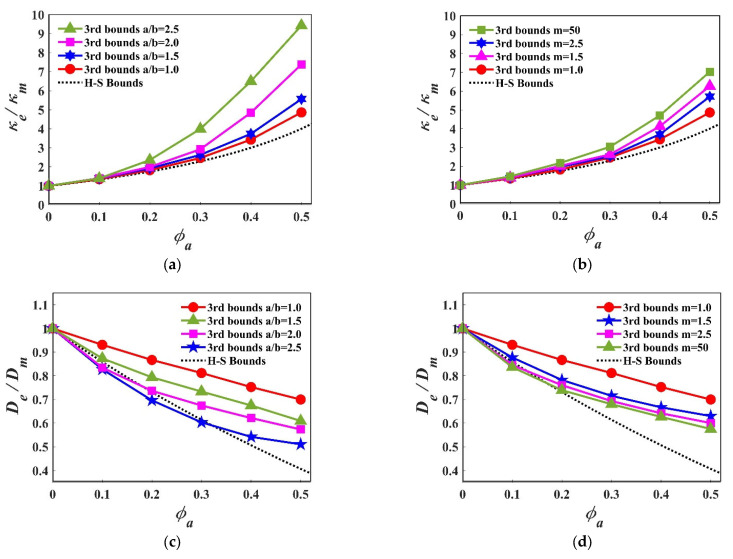
Third-order approximation of effective properties with strong contrast: (**a**) Normalized thermal conductivity with different aspect ratio; (**b**) Normalized thermal conductivity with different deformation coefficient; (**c**) Normalized diffusion coefficient with different aspect ratio; (**d**) Normalized diffusion coefficient with different deformation coefficient.

**Table 1 materials-15-05799-t001:** Total consumption times of the GPU and CPU on packing system of spheres.

	$S_{a}$	$S_{aa}$	$S_{aaa}$
GPU	0.12 s	14.63 s	101,679.53 s (≈28 h)
CPU	12.96 s	1057.95 s	9,691,950.31 s (≈2139 h)

Note: packing system and discrete number of samplings are closely related to the execution time. The parameters in this test are shown as follows: the packing fraction $\phi_{a}=0.5$ and total particle number $N_{p}=954$; the sampling points/lines/triangles of $S_{a}$/$S_{aa}$/$S_{aaa}$: $N_{sampling}=2\times 10^{6}$; the discrete values of $S_{aa}$/$S_{aaa}$ are $N_{r}=100$ and $N_{r_{1}}\times N_{r_{2}}\times N_{\theta }=100\times 100\times 100$, respectively.

**Table 2 materials-15-05799-t002:** A fraction of super-ellipsoids with particular range.

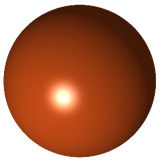 a/b=1 m=1	a/b=1.5, m=1, b=c	a/b=2,m=1, b=c	a/b=2.5,m=1, b=c
	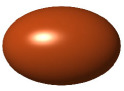	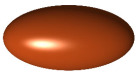	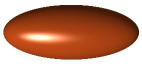
	m=1.5, a=b=c	m=2.5,a=b=c	m=50,a=b=c
	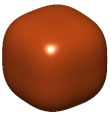	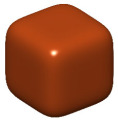	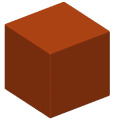

## Data Availability

Not applicable.

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
