# Peer review of "Third-Order Effective Properties for Random-Packing Systems Using Statistical Micromechanics Based on a GPU Parallel Algorithm in Fast Computing n-Point Correlation Functions"

_materials, 2022, doi:10.3390/ma15165799_

Round 1

Reviewer 1 Report

The article presents GPU based parallel algorithm to estimate third-order effective properties including thermal conductivity and diffusion coefficient. Verifications are conducted and show the quality of the results.

The article is very interesting and totally according with the scope of the journal.

Although the main focus of the article is Material science, as it presents a GPU parallel algorithm I think the authors should presents the execution times so, other researches can compare with other solutions or have an idea if they could use to solve their problems.

My suggestion is that the author add a table in the article with the sequential (if feasible) an parallel times for a few instances.

Some small corrections:

Section 3

                Between section 3 and subsection 3.1 briefly explain what will be presented in the section.

Line 191 I did not understand the following sentence “However, very computer has a very limited 191 CPU cores”. Specifically, I did not understand the word “very”. You mean “every”?

It seems figure 4 outside the margins

Reviewer 2 Report

1. The article presents exciting and practical solutions to estimating the effective properties of the particulate system.

2. The literature review introduces the reader to the issue in a correct way, and explains the evolution of solutions. However, it is worth indicating their limitations in a more precise way. Most of the cited works come from quite distant years, their scientific value may certainly have gotten out of date. If possible, please update your references.

3. The presented equations are correct and understandable. The methodology is also described professionally, concisely and clearly.

4. No objections to the evaluation of solutions.

5. The work requires many editorial corrections. You can find an excess of semicolons in this article. Oftentimes, the correct introduction to the equation is missing. Incorrect punctuation also appears in the equations. I made a note of it throughout the whole article. In descriptions of figures, sentences end differently with a semicolon or a dot or none. Please unify in reference to editorial guidelines.

6. Figure 7 should be on one page together with the description.

7. Citing the article "et al." is used, not "et al". It is also good practice to add the year of publication.
